# Fast and reversible neural inactivation in macaque cortex by optogenetic stimulation of GABAergic neurons

Abhishek De[1,2], Yasmine El-Shamayleh[3], Gregory D Horwitz[2]*

[1]Graduate Program in Neuroscience, University of Washington, Seattle, United States; [2]Department of Physiology and Biophysics, Washington National Primate Research Center, University of Washington, Seattle, United States; [3]Department of Neuroscience, Zuckerman Mind Brain Behavior Institute, Columbia University, New York, United States

**Abstract** Optogenetic techniques for neural inactivation are valuable for linking neural activity to behavior but they have serious limitations in macaques. To achieve powerful and temporally precise neural inactivation, we used an adeno-associated viral (AAV) vector carrying the channelrhodopsin-2 gene under the control of a Dlx5/6 enhancer, which restricts expression to GABAergic neurons. We tested this approach in the primary visual cortex, an area where neural inactivation leads to interpretable behavioral deficits. Optical stimulation modulated spiking activity and reduced visual sensitivity profoundly in the region of space represented by the stimulated neurons. Rebound firing, which can have unwanted effects on neural circuits following inactivation, was not observed, and the efficacy of the optogenetic manipulation on behavior was maintained across >1000 trials. We conclude that this inhibitory cell-type-specific optogenetic approach is a powerful and spatiotemporally precise neural inactivation tool with broad utility for probing the functional contributions of cortical activity in macaques.

*For correspondence:
ghorwitz@uw.edu

Competing interests: The authors declare that no competing interests exist.

## Introduction

A major goal of systems neuroscience is to understand how neural activity mediates behavior. Neural inactivation techniques are central to this endeavor (*Wurtz, 2015*). However, these techniques can have unintended consequences that complicate data interpretation (*Abraham, 2008*; *Goold and Nicoll, 2010*; *Goshen et al., 2011*; *Sokolova and Mody, 2008*; *Stemmler and Koch, 1999*; *Turrigiano et al., 1998*). For example, by impairing task performance, neural inactivation can cause animals to explore new task strategies for acquiring reward. This change in strategy may change the information flow through neural circuits. To avoid these complications, inactivation methods are needed that can be reversed more quickly than these circuit-level changes can occur.

Optogenetics is the fastest method for reversible neural inactivation currently available. In rodents, optogenetic inactivation has revealed links between neural activity and behavior that would have been difficult to discover with traditional, slower inactivation methods based on injection of pharmacological agents, cortical cooling, or lesioning (*Goshen et al., 2011*; *Hanks et al., 2015*; *Yartsev et al., 2018*). Optogenetic inactivation has already been used in a few pioneering studies to perturb the behavior of macaque monkeys (*Acker et al., 2016*; *Afraz et al., 2015*; *Cavanaugh et al., 2012*; *Fetsch et al., 2018*). The approach taken in these studies was to reduce neuronal spiking by activating hyperpolarizing opsins (eNpHR, Arch, or Jaws). The directness of this approach facilitates the interpretation of behavioral effects. However, the behavioral effects produced this way have been small, perhaps because most promoters used in viral vectors drive expression in many neuronal types, and suppression of inhibitory neurons may counteract suppression of excitatory neurons.

An alternative approach, which has been successful in rodents, is to selectively activate inhibitory neurons with channelrhodopsin-2 (ChR2) (*Cone et al., 2019*; *Glickfeld et al., 2013*; *Guo et al., 2014*; *Khan et al., 2018*; *McBride et al., 2019*). This approach has two advantages. First, it is based on the opening of ion channels, which conduct more ions per photon absorbed than ion pumps. Second, it leverages the dense local connectivity and low synaptic failure rates of GABAergic neurons to suppress long-range excitatory signaling locally and robustly (*Isaacson and Scanziani, 2011*; *Kubota et al., 2015*; *Packer and Yuste, 2011*; *Wiegert et al., 2017*).

To test the efficacy of this approach for cortical inactivation in macaques, we injected area V1 of three rhesus monkeys with a viral vector containing a cell-type-specific promoter (AAV–mDlx5/6–ChR2) (*Dimidschstein et al., 2016*). In this study, we confirm the specificity of GABAergic neuronal transduction in macaque cortex and demonstrate that illumination of the injection site modulates spiking activity. We also show that illumination impairs visual sensitivity profoundly, reversibly, and reliably at the receptive fields of the illuminated neurons but not outside. We conclude that optogenetic stimulation of inhibitory neurons is a powerful method for inactivating regions of the macaque monkey brain with high spatial and temporal precision.

## Results

### Selectivity of opsin expression

A previous study showed that an AAV vector carrying the gene for the fluorescent reporter, GFP, under the control of the mDlx5/6 enhancer, transduced V1 GABAergic neurons in a marmoset with 93% selectivity (*Dimidschstein et al., 2016*). To determine whether AAV–mDlx5/6–ChR2–mCherry has similar selectivity in macaque, we injected V1 of one animal (monkey 1) and examined the tissue histologically (*Figure 1* and *Figure 1—figure supplement 1*). mCherry-positive cells had non-

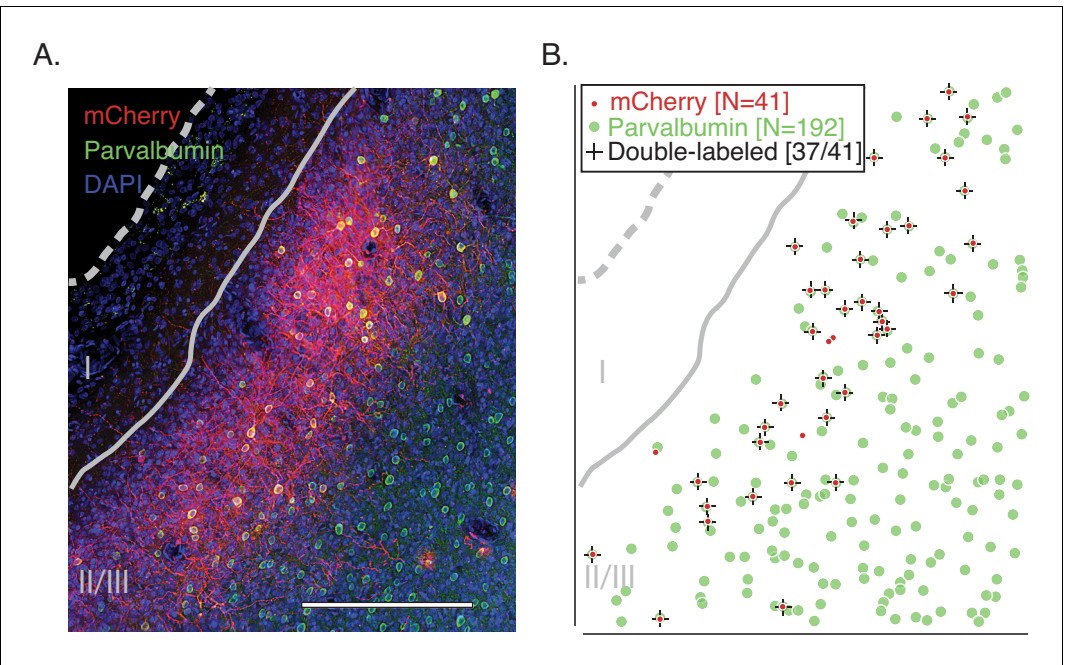

**Figure 1.** Immunohistochemical analysis of transduction by AAV1-mDlx5/6-ChR2-mCherry. (**A**) A histological section of V1 from monkey 1 stained with DAPI (blue) and antibodies against parvalbumin (green) and mCherry (red). Scale bar is 250 µm. The pial surface is indicated by the dashed gray curve and the border between layers 1 and 2/3 is indicated by the solid gray curve. The laminar specificity is an idiosyncrasy of this particular injection; see *Figure 1—figure supplement 1* for a histological section of the V1/V2 border. (**B**) Locations of cell bodies in (**A**) expressing mCherry (red), parvalbumin (green), or both ('+').

The online version of this article includes the following figure supplement(s) for figure 1:

**Figure supplement 1.** Immunohistochemical analysis of transduction by AAV1-mDlx5/6-ChR2-mCherry.

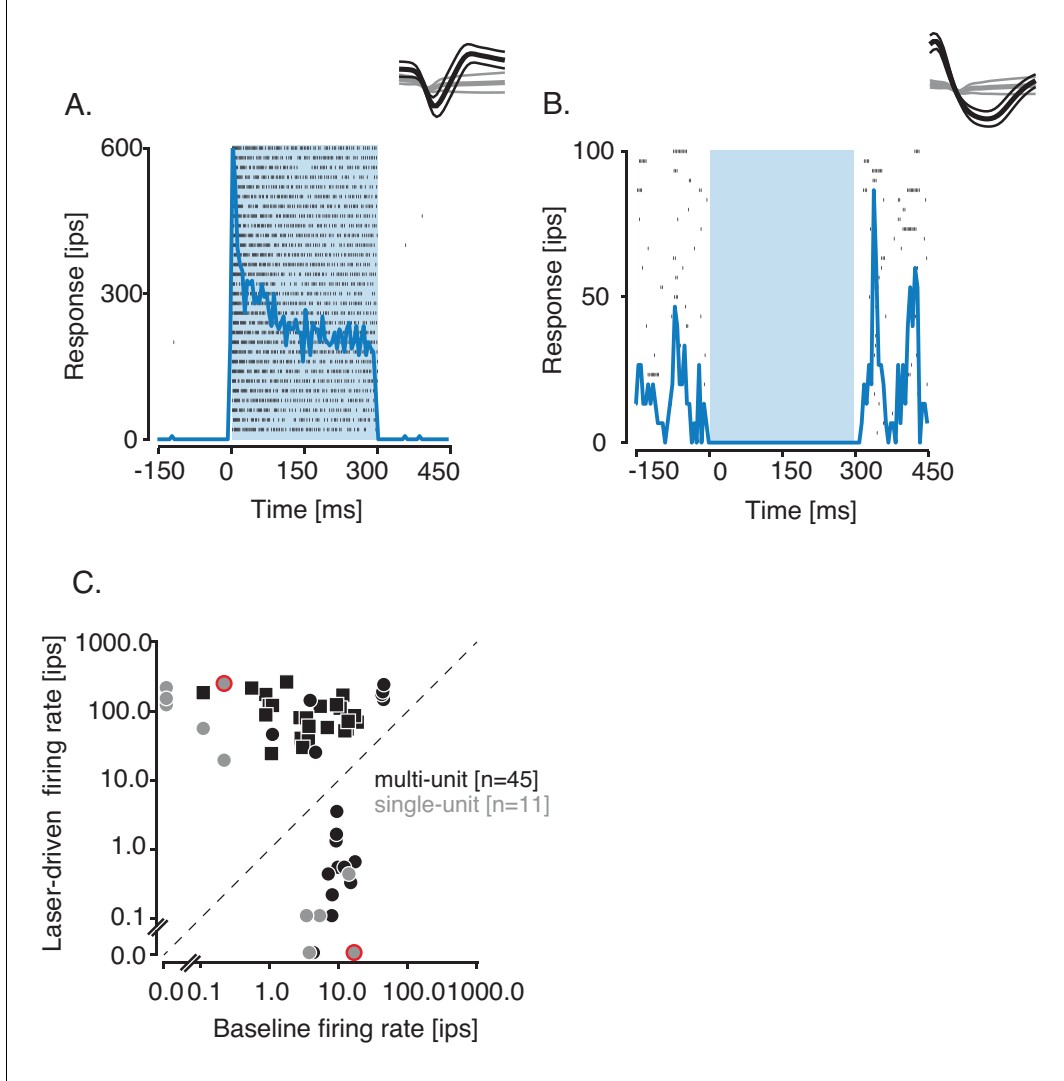

**Figure 2.** Optogenetic activation and suppression of single- and multi-units. (A,B) Responses (in impulses per second, ips) of two example single units, aligned to the onset of optical stimulation, which lasted 300 ms (blue rectangle). Rasters (tick marks) and peristimulus time histograms (blue traces) are shown for an activated single unit (A) and a suppressed single unit (B). Insets: Mean spike waveform (thick black curve) and noise waveform (thick gray curve) ± 1 standard deviation (thin curves). (C) Scatter plot of firing rate on laser trials against baseline firing rate of units from monkey 2 (squares) and monkey 3 (circles). Data from example activated and suppressed units are circled in red. Firing rates were computed during optical stimulation or the equivalent epoch on control trials. The online version of this article includes the following figure supplement(s) for figure 2:

**Figure supplement 1.** Analysis of visually driven responses at activated and suppressed sites.
**Figure supplement 2.** Analysis of latency at activated and suppressed sites.
**Figure supplement 3.** Rasters from all of the 18 suppressed sites.

pyramidal morphologies, consistent with them being GABAergic. Similar histological results with this viral vector have been described in macaques previously (*Scerra et al., 2019*).

Most mCherry-positive neurons co-expressed parvalbumin (468/543), a marker for 75% of GABAergic neurons in macaque V1 (*Van Brederode et al., 1990*). This high level of co-expression is consistent with selective transduction of GABAergic neurons and is sufficiently high to suggest that parvalbumin-positive neurons were transduced with particularly high efficiency (p<0.005; binomial test).

## Optogenetic control of neural activity

To test whether ChR2 expression was sufficiently strong to perturb neural activity, we recorded extracellular spiking responses from single- and multi-units near the injection sites in two other monkeys (monkeys 2 and 3) while they performed a contrast detection task. Most sites were visually driven (46/56, response to a low-contrast Gabor stimulus greater than baseline firing rate; 19/56, p<0.05; Mann-Whitney U test, *Figure 2—figure supplement 1*). Given our selection criteria, all sites were significantly modulated by optical stimulation (p<0.06; Mann-Whitney U test; see Methods). Some units were excited by optical stimulation (*Figure 2A*) whereas others were suppressed (*Figure 2B*). At 38 of the 56 sites, optical stimulation increased spiking. Excitation was prevalent in our dataset because we searched for sites at which optical stimulation produced an audible change in the baseline firing rate (*Figure 2C*). The mean latency to response was 14±26 (SD) ms and was <5 ms at 11 sites (*Figure 2—figure supplement 2A–B*). Neurons excited at short latency (<5 ms)

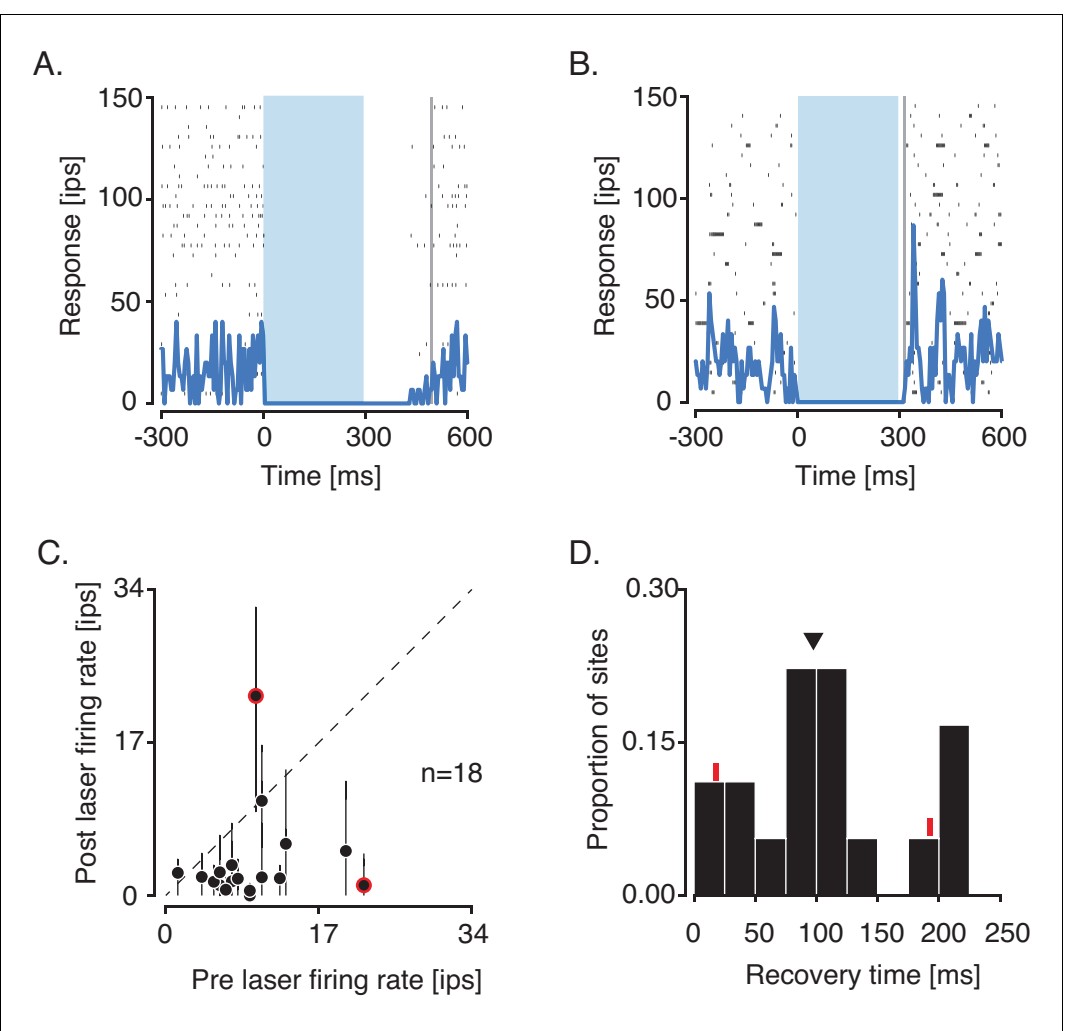

**Figure 3.** Analysis of activity rebound and recovery at suppressed sites. (A) Responses of a single unit with a pre-laser firing rate that exceeded the post-laser firing rate. Recovery time to the baseline firing rate was 195 ms (vertical gray line). (B) Responses of another single unit with a post-laser firing rate that exceeded the pre-laser firing rate. Recovery time was 15 ms. (C) Scatter plot of pre-laser firing rates against post-laser firing rates. For each site, post-laser firing rate was computed in a sliding 50-ms window from 0 to 200 ms after the laser was switched off. The ranges of post-laser firing rates are plotted as black lines and averages are plotted as black points. Data from neurons in (A) and (B) are circled in red. (D) Histogram of recovery times following optogenetic suppression. Recovery times of example units are marked with red tick marks, and the median is marked with a triangle.

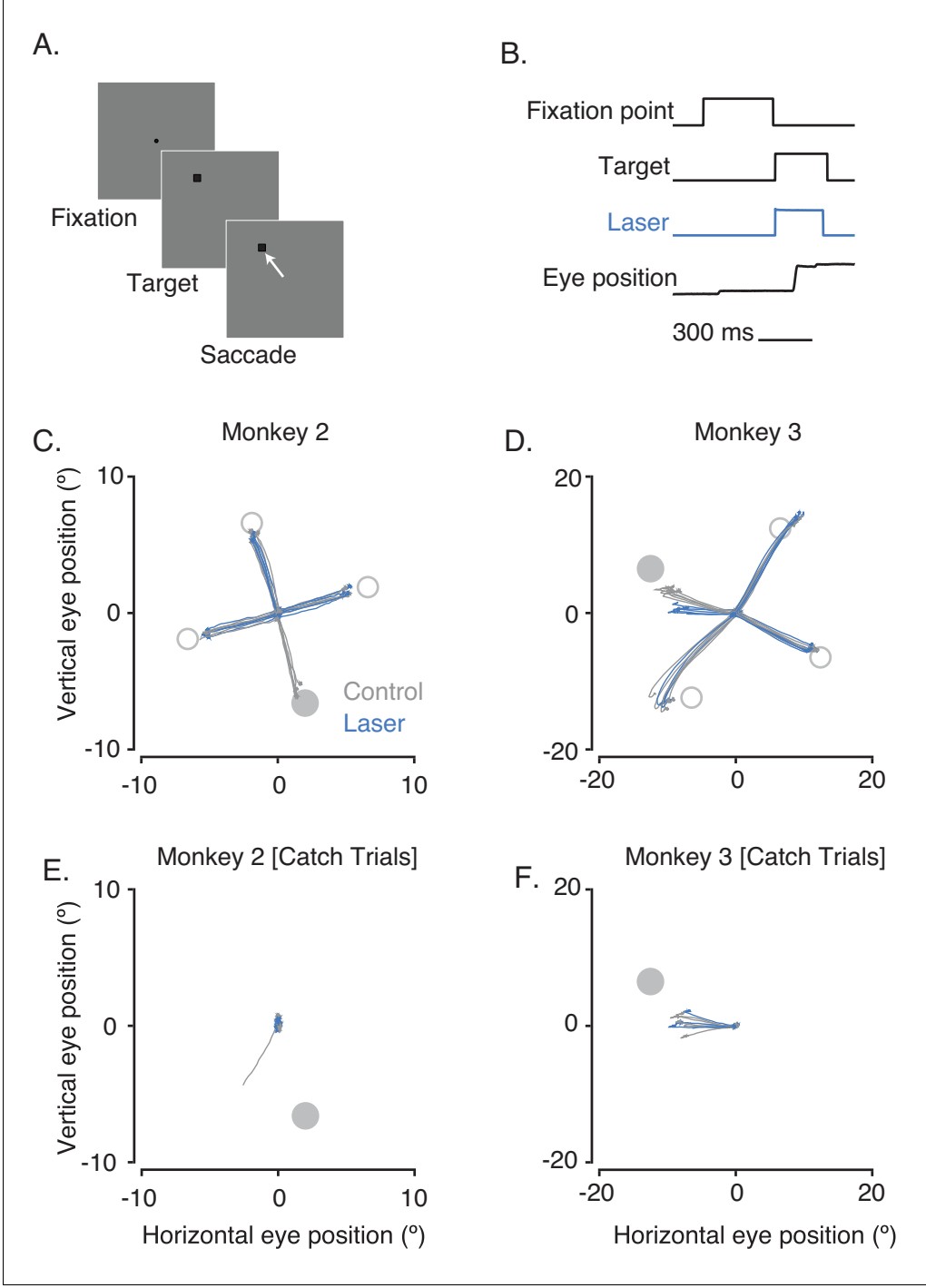

**Figure 4.** Effect of optogenetic inactivation of V1 on visually guided saccades. (A) Task design. (B) Timing of events. The small overshoot in the laser trace accurately reflects the temporal profile of the light. (C,D) Eye position traces on control (gray) and laser (blue) trials are shown from 0 to 300 ms after the fixation point was extinguished for one block of trials. The RF location of the illuminated neurons (gray filled circle) and the target locations outside of the RF (gray open circles) are highlighted. (E,F) Eye positions on catch trials from the same blocks as (C,D).

The online version of this article includes the following figure supplement(s) for figure 4:

**Figure supplement 1.** Effect of optogenetic inactivation on visually guided saccades from monkey 3.

**Figure supplement 2.** Effects of optogenetic inactivation of V1 on visually guided saccade accuracy and latency.

presumably expressed ChR2 and suppressed other neurons via synaptic inhibition. The latency of suppression was longer than the latency of excitation, but this comparison is challenging because baseline firing rates were low (*Figure 2—figure supplement 2C*, *Figure 2—figure supplement 3*).

Neural activity suppression using halorhodopsins in monkeys is typically followed by a rebound of activity at the termination of optical stimulation (*Acker et al., 2016*; *Fetsch et al., 2018*). We did not observe such rebounds with AAV–mDlx5/6–ChR2. We compared average firing rates at 18 suppressed sites in a 50-ms window before and after optical stimulation. At one example site, the pre-laser firing rate exceeded the post-laser firing rate (22 vs. 0 impulses/sec, p<0.001, Wilcoxon signed rank test, *Figure 3A*), consistent with sustained suppression. At a different example site, the pre-laser firing rate was lower than the post-laser firing rate, consistent with a small rebound (10 vs. 25 impulses/sec, p=0.02, Wilcoxon signed rank test, *Figure 3B*). Such rebounds were rare; post-laser firing rates exceeded pre-laser firing rates at only 2 of 18 sites (*Figure 3C*).

Activity at most suppressed sites recovered to baseline levels gradually after laser termination. We measured this recovery time by computing the first time at which the average spike count in a 50-ms sliding window returned to 90% of the pre-laser firing rate. Recovery times ranged from 0 to 215 ms (*Figure 3D*) with roughly half of the sites recovering within 100 ms (median = 97.5 ms). These data demonstrate that suppression persists several tens of milliseconds after laser termination.

## Optogenetic control of behavior

To evaluate the behavioral efficacy of optogenetic stimulation, we trained monkeys 2 and 3 to perform two visually demanding tasks. Reward contingencies were independent of laser stimulation in both tasks.

In the visually guided saccade task, a target appeared inside the receptive fields (RFs) of the stimulated V1 neurons on a random subset of trials and outside on other trials (*Figure 4A–B*). Data from an example block of trials from each monkey show the main results (*Figure 4C–D*). On control trials, both monkeys made accurate saccades to most target locations. On laser trials, the monkeys failed to make saccades into the RFs of the optically stimulated neurons. Saccades were unaffected when the target appeared at other locations, indicating that the optogenetic effect was retinotopically specific. On laser trials when the target appeared inside the RFs, monkey 2 typically maintained fixation, and monkey 3 typically made leftward ~10° saccades. Similar behaviors were observed on catch trials in which no target was shown (*Figure 4E–F*, see Materials and methods). The inaccuracy of saccades made by monkey 3 into the left visual field was likely due to repeated electrode penetrations in the midbrain of this animal that were unrelated to the current experiments (*Figure 4—figure supplement 1*).

We collected data in 10 sessions from monkey 2 (16 blocks of trials) and 7 sessions from monkey 3 (20 blocks of trials). Within each session, we calculated the distance between saccade end points and target locations. When the target appeared inside the RFs of stimulated neurons, the saccade end points tended to be closer to the target on control trials than on laser trials (p<0.002 for monkey 2, p=0.03 for monkey 3; Wilcoxon signed rank tests). When the target appeared in other locations, the saccade endpoints were similarly close to the target on control and laser trials (p=0.92 for monkey 2, p=0.38 for monkey 3; Wilcoxon signed rank tests; *Figure 4—figure supplement 2A–B*). Saccade latencies were greater on laser trials than on control trials when targets were inside the RFs (p<0.0001 for monkey 2 and 3; Mann-Whitney U tests; *Figure 4—figure supplement 2C–D*) but not when targets were elsewhere (p=0.90 for monkey 2 and p=0.41 for monkey 3; Mann-Whitney U tests; *Figure 4—figure supplement 2E–F*).

To confirm that the deficit in task performance was not purely oculomotor, we trained monkeys 2 and 3 to perform a contrast detection task that required saccades to targets outside of the RFs of the stimulated neurons (*Figure 5A–B*: see Materials and methods). An example block of trials from each monkey demonstrates the main results. Both monkeys detected the visual stimulus more frequently on control than on laser trials (proportion of hits on control vs. laser trials; p<0.001 for monkey 2 and monkey 3; binomial tests for equality of proportions; *Figure 5C–D*). This performance deficit was also reflected in psychometric functions (*Figure 5C–D* inset) and in the contrast values selected by the staircase procedure (*Figure 5E–F*). Neither monkey was able to detect the visual stimulus with above-chance accuracy on laser trials even at the maximum stimulus contrast achievable. Saccades to the stimulus location were never required, and thus the behavioral effects produced by optical stimulation in this task cannot be explained by an oculomotor deficit.

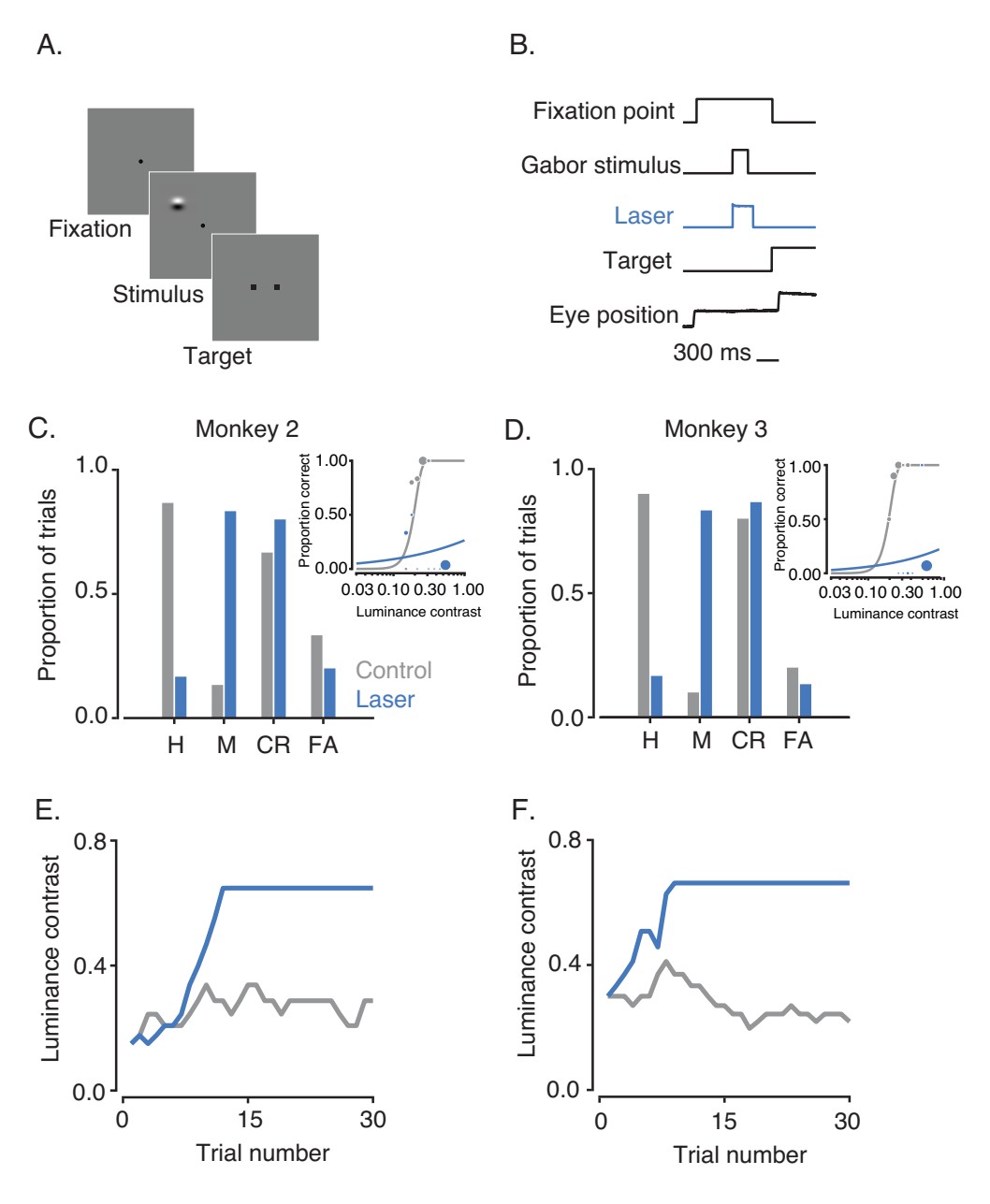

**Figure 5.** Effect of V1 inactivation on visual contrast detection. (**A**) Task design. (**B**) Timing of events. (**C**) Performance of monkey 2 over one block of trials. Hits (H) and misses (M) are proportions of Gabor-present trials that were answered correctly and incorrectly, respectively. Correct rejections (CR) and false alarms (FA) are proportions of Gabor-absent trials that were answered correctly and incorrectly, respectively. Insets show psychometric functions on control (gray) and laser (blue) trials. Symbol size in insets reflects the number of trials that contributed to each data point. (**E**) Contrasts selected by the staircase procedure on control (gray) and laser (blue) trials. (**D,F**) Performance of monkey 3 in the same format as (**C,E**). Luminance contrast could not exceed 0.66 because the gray background was close to the upper limit of the display range.

In one session, the Gabor stimulus location was randomized across trials, confirming the retino-topic specificity of the effect (*Figure 6*). Additional control experiments confirmed that the monkeys were able to make saccades to both target locations irrespective of optical stimulation (data not shown) and showed that performance on control trials was unaffected by the interleaved laser trials (*Figure 6—figure supplement 1*).

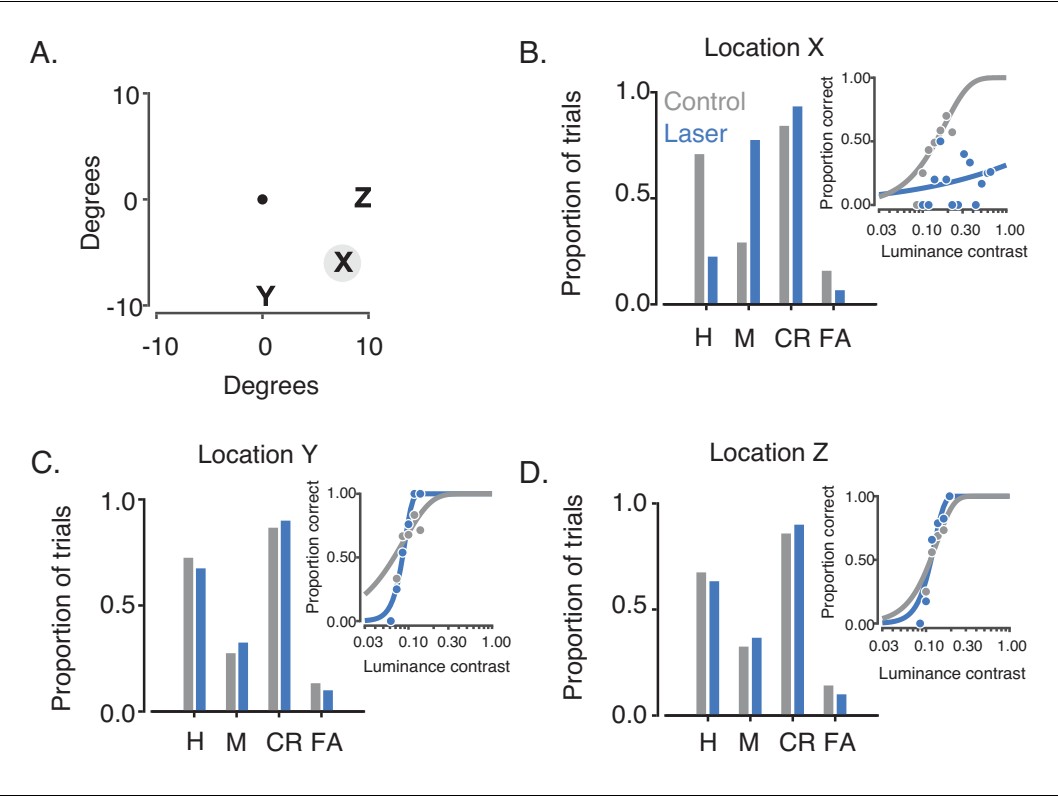

**Figure 6.** Retinotopic specificity of optogenetic effects on contrast detection. Data are from a single session consisting of 4 blocks of trials from monkey 2. (**A**) On each trial, the Gabor stimulus appeared at one of three randomly interleaved locations (X, Y, or Z), all of which were 9.6° away from the fixation point (central black dot). Locations Y and Z were on the vertical and horizontal meridians, respectively. (**B**) The proportions of hits (H), misses (M), correct rejections (CR) and false alarms (FA) are plotted in the same format as in *Figure 5C*. The laser reduced the monkey's contrast sensitivity when the Gabor stimulus appeared at the receptive fields of the transduced neurons (X, gray circle). No significant effect was observed at locations Y and Z (**C,D**). The online version of this article includes the following figure supplement(s) for figure 6:

**Figure supplement 1.** Contrast detection thresholds were stable across seven blocks collected during a single session from monkey 3.

We collected data in 11 sessions from monkey 2 (69 blocks of trials) and 12 sessions from monkey 3 (81 blocks of trials). In almost every session (10/11 in monkey 2, 11/12 in monkey 3), the proportion of hits on control trials was significantly greater than on laser trials (binomial tests for equality of proportions, p<0.05, *Figure 7A–B*). An analysis of sensitivity indices (*d′*) confirmed that this change in performance was consistent with a reduction in sensitivity and inconsistent with a pure change in criterion (*Figure 7C–D*, *Figure 7—figure supplement 1*). In most blocks of trials (52/69 in monkey 2 and 63/81 in monkey 3), optical stimulation increased detection thresholds beyond the limits of the display, an event that occurred rarely on control trials (0/63 blocks in monkey 2, 8/81 blocks in monkey 3).

As laser power increased, errors became more common, which caused the staircase procedure to increase the stimulus contrast rapidly (*Figure 8A*). The magnitude of the behavioral effect increased steeply with laser intensity between 12.8 and 22.3 mW, and it saturated by 30.0 mW (*Figure 8B*). Behavior on control trials was not significantly affected by changes in laser power (r=−0.15, p=0.78; Spearman's correlation between *d′* on control trials from each block and laser power).

Optogenetic modulations of neural activity were linked to effects on behavior across these trials (r=0.36, p=0.43; Spearman's correlation between neural laser modulation index and difference in *d′* between control and laser trials; *Figure 8—figure supplement 1*). Pooling data across all blocks of trials reduced the correlation (r=0.16, p=0.23). Pooling the data reduced statistical power due to covariates across blocks of trials that exerted different effects on neurophysiological and behavioral

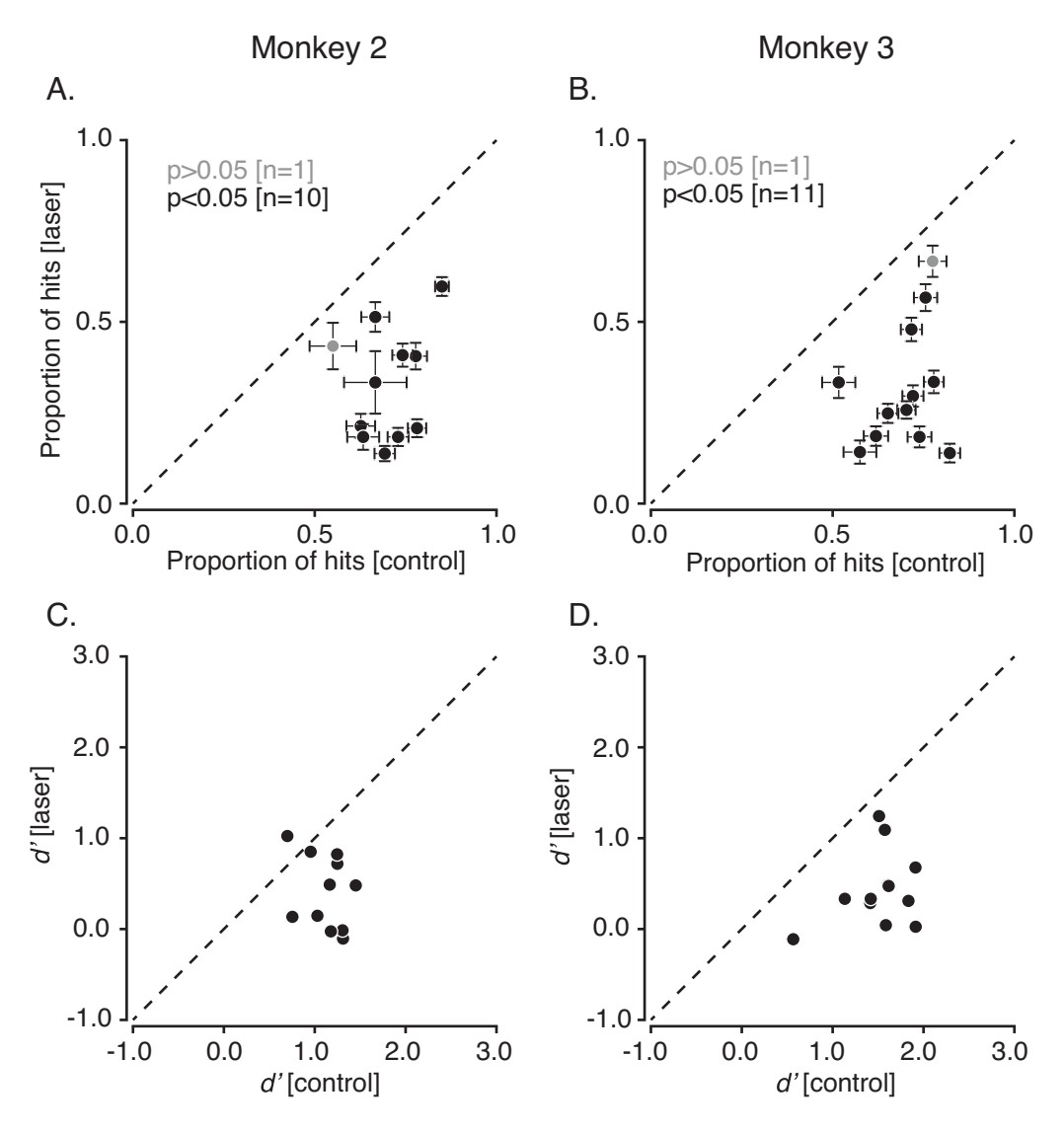

**Figure 7.** Effect of V1 inactivation on visual contrast detection across multiple sessions. (A) Scatter plot showing proportion of hits on control trials against laser trials from each session in monkey 2. Sessions with significantly fewer hits on laser trials than control trials are shown in black (p<0.05, binomial test for equal proportions). Error bars represent the standard error of mean. (B) Data from monkey 3 in the same format as (A). (C) Scatterplot of *d'* from control trials plotted against *d'* from laser trials from each session performed by monkey 2. (D) Data from monkey 3 in the same format as (C).

The online version of this article includes the following figure supplement(s) for figure 7:

**Figure supplement 1.** Analysis of the relationship between *d'* and c-criterion.

outcomes (e.g. fiber position, stimulus location in the visual field, and quality of neural recordings). These covariates were held fixed in the data shown in *Figure 8A–B*.

In a previous study, the behavioral effects produced by optogenetic silencing of neurons in area MT using the suppressive opsin, Jaws (red-shifted cruxhalorhospsin), decreased over tens of minutes (*Fetsch et al., 2018*). To determine whether a similar phenomenon occurred with ChR2-mediated inactivation, we analyzed 4 experimental sessions (28 blocks of trials) from monkey 2 and 5 sessions (33 blocks of trials) from monkey 3. From these sessions, we considered only the subset of blocks with identical laser power.

The behavioral effects we observed were consistently large over the course of ~1000 trials (or ~50 mins). We quantified the behavioral effect as the difference in $d'$ between laser and control trials within each block. The behavioral effect varied little as a function of block number, (r=0.18, p=0.59; Spearman's correlation between block number and $d'$ averaged across sessions; *Figure 8C*). It was also consistent within individual sessions; linear regression slopes of the behavioral effect as a function of block number in each session did not differ significantly from zero (p=0.57, Student's t-test). For comparison with previous work (*Fetsch et al., 2018*), we calculated the behavioral effect in early and late trials within each session. Unlike the previous work, the behavioral effect did not differ between the first 480 trials (4 blocks) and the subsequent trials, suggesting the absence of compensatory changes under the conditions of the current study (p=0.79, Student's t-test, *Figure 8D*).

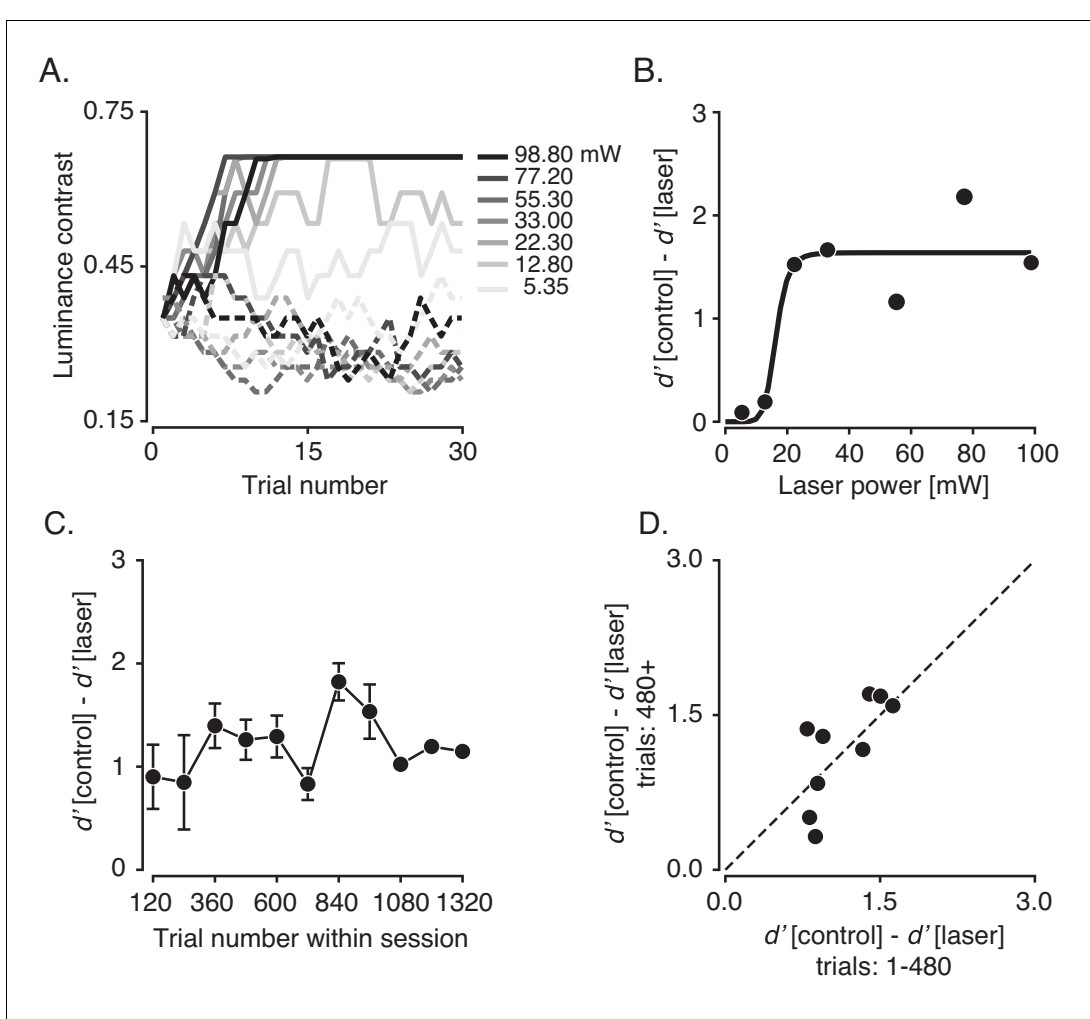

**Figure 8.** Effect of laser power and repeated optical stimulation on contrast detection. (**A**) Contrast values selected by the staircase procedure on laser trials (solid lines) and interleaved control trials (dashed lines) across seven blocks. (**B**) The difference in $d'$ between control and laser trials as a function of laser power calculated from the data in (**A**). A Naka-Rushton fit to the data is shown in black. (**C**) Differences in $d'$ between control and laser trials as a function of trial number in each session. Each session consisted of at least five blocks of 120 trials. The duration of an individual trial was 2.80 ± 0.51 s (mean ± SD), and the number of trials per session was 813 ± 253. Points are means and error bars are standard error of the mean (SEM). SEM was not plotted for the final two points, each of which represent data from a single session. (**D**) Scatter plot of the differences in $d'$ for early trials (1–480) vs. late trials (480–beyond) within each session.

The online version of this article includes the following figure supplement(s) for figure 8:

**Figure supplement 1.** Correlation between optogenetic effects on neural activity and behavior.

**Figure supplement 2.** Analysis of visual sensitivity in monkey 2.

## Discussion

The fast activation and inactivation of neurons afforded by optogenetics has revolutionized our understanding of the nervous systems of rodents and invertebrates. Understanding the primate brain at a similar level of detail is facilitated by optogenetics in the macaque monkey—a model organism with a brain structure similar to humans that can be trained to perform complex behavioral tasks. Rapid activation was already feasible in primates using microsimulation or optogenetics, and now rapid inactivation is too.

We achieved inactivation by stimulating GABAergic neurons in macaque V1 and measured electrophysiological and behavioral consequences. First, we showed that the AAV–mDlx5/6–ChR2 vector targeted ChR2 expression to GABAergic neurons in area V1. Second, we showed that optical stimulation modulated the activity of neurons near the injection site. Third, we showed that optical stimulation impaired visual sensitivity in two behavioral tasks. The reduction in sensitivity was specific to trials in which optical stimulation was delivered and to the RF location of the stimulated neurons, demonstrating the temporal and spatial precision of the inactivation. Laser-induced modulations of neural responses were rapid, rebound activity following light pulses was negligible, and behavioral effects were consistent across ~1000 trials.

Below, we compare the results of our study with those of previous studies that used optogenetic neural inactivation to perturb macaque behavior. We then discuss the effect of optogenetic stimulation of V1 on eye movements and ways in which the method could be improved. Finally, we discuss potential applications of AAV–mDlx5/6–ChR2 for understanding primate brain function.

### Comparison with optogenetic inactivation studies

Four previous studies used optogenetic inactivation to perturb monkey behavior (*Acker et al., 2016*; *Afraz et al., 2015*; *Cavanaugh et al., 2012*; *Fetsch et al., 2018*). The two studies most similar to ours quantified the effect of neural inactivation on behavior as changes in visual discrimination thresholds on 2AFC tasks (*Afraz et al., 2015*; *Fetsch et al., 2018*). In one study, inactivation of inferotemporal cortical neurons raised thresholds for classifying face stimuli on the basis of gender (*Afraz et al., 2015*). In the other, inactivation of MT cortical neurons biased judgements of visual motion direction (*Fetsch et al., 2018*). In both cases, threshold changes were smaller than those we observed (~5% vs. >100%, *Figure 7*).

The threshold changes we observed were large for potentially several reasons. First, we excited ChR2-expressing inhibitory neurons to reduce the spiking of excitatory neurons. Stimulation of a small number of inhibitory neurons can suppress the activity of a large number of excitatory neurons (*Wiegert et al., 2017*). Second, we used ChR2, which conducts more ions per absorbed photon than ion pumps (Jaws and ArchT). Third, we used higher laser power (4–160 mW vs. ~2 mW and ~12 mW; *Figures 4–8*). Fourth, we inactivated area V1, an area that is indispensable for the behaviors we studied (*Koerner and Teuber, 1973*; *Merigan et al., 1993*; *Radoeva et al., 2008*). Higher-order visual cortical areas may be sufficiently interconnected to allow one or more areas to compensate for others with regard to the behaviors tested. An intriguing, and now-testable, hypothesis is that the spared visual sensitivity following visual cortical lesions is due to the engagement of slow compensatory mechanisms, not the unmasking of normally functioning pathways (*Leopold, 2012*).

We interpreted the stimulation-induced change in the monkeys' performance as a change in sensitivity, and it is inconsistent with a change in criterion alone. Additionally, the brevity and unpredictability of the optical stimulation makes large, consistent, selective changes in criterion on laser trials unlikely. Nevertheless, we cannot rule out the possibility that the optical stimulation affected sensitivity and criterion together (*Figure 7—figure supplement 1*).

### Comparison with non-selective optogenetic stimulation of V1

Illumination of ChR2-expressing neurons in area V1 causes monkeys to make saccades into the RFs of the stimulated neurons under some conditions (*Jazayeri et al., 2012*). This behavior is consistent with the perception of a phosphene (*Tehovnik et al., 2003*). In our study, however, monkeys rarely made saccades into the RFs of the stimulated neurons, suggesting that they did not experience a phosphene. This result held on trials requiring a saccade to a visual target inside the RFs of the stimulated neurons and on trials in which no target was shown, a condition similar to the key condition in the study of *Jazayeri et al., 2012*. In principle, detection of the optical stimulation could have

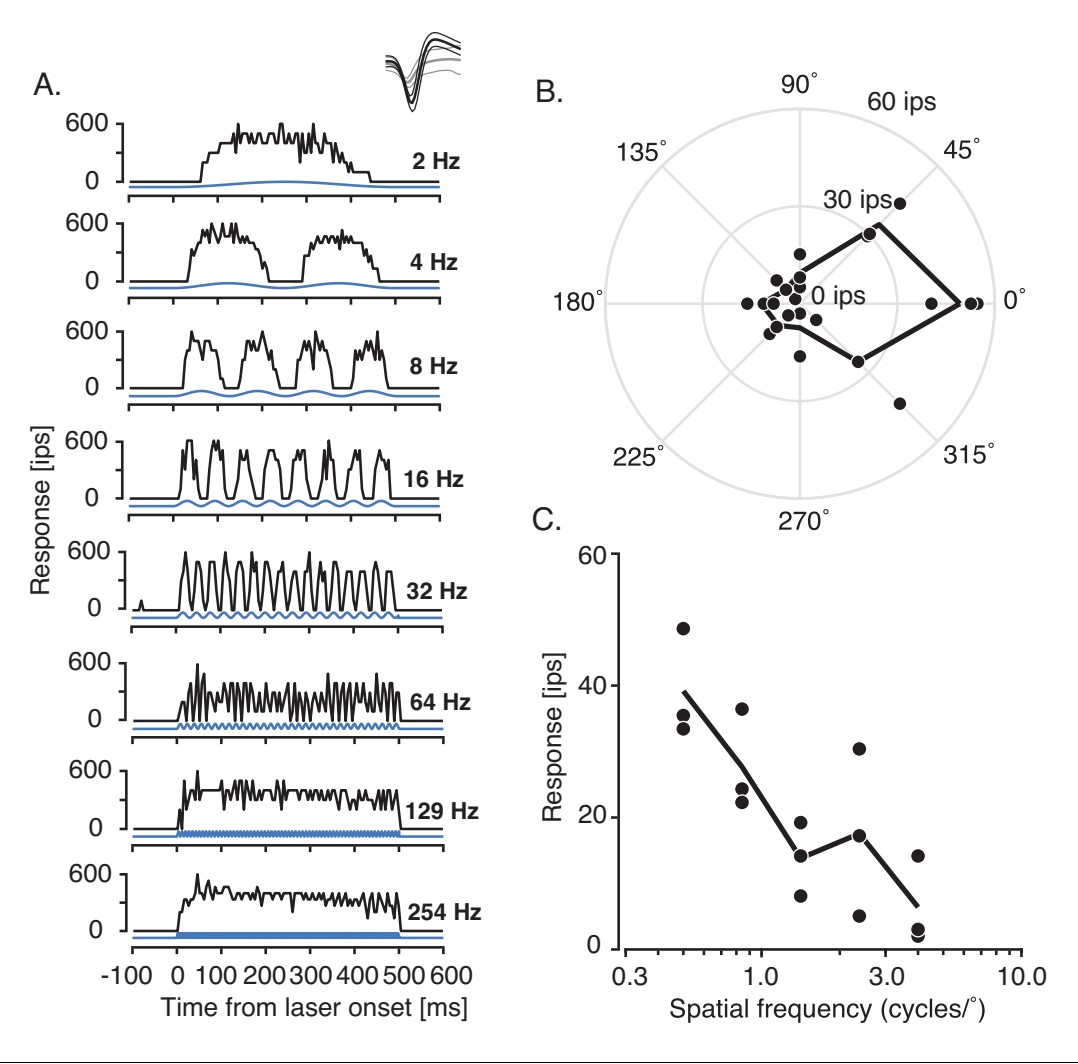

**Figure 9.** Responses of a putative GABAergic, direction-selective single unit to optical (450 nm laser) and visual (drifting achromatic 3 Hz sinusoidal grating) stimulation (**A**). Peristimulus time histograms (black) in response to sinusoidal laser modulation from 2 Hz to 254 Hz (blue). Inset: Mean spike waveforms (thick black curve) and noise waveforms (thick gray curve) ± 1 standard deviation (thin curves). (**B**) Direction tuning curve showing individual (black points) and mean responses (black line) across repeated presentations of a drifting sinusoidal grating. (**C**) Spatial frequency tuning curve with symbols identical to (**B**).

provided a cue for acquiring reward in the visually guided saccade task. Having sensed the optical stimulation, and seen no target, the monkey could have increased its reward rate by making a saccade into the RFs of the stimulated neurons. The fact that the monkeys did not behave this way suggests that they were unable to detect the stimulation, or at least were unable to use it to direct saccades. Sensing the optical stimulation would not have been useful for increasing reward rate in the contrast detection task.

We attribute the difference between studies to the population of V1 neurons stimulated. We used a Dlx5/6 enhancer to express ChR2 selectively in inhibitory neurons whereas *Jazayeri et al., 2012* used the human synapsin I promoter, which drives expression in both excitatory and inhibitory neurons (*Nathanson et al., 2009*). One hypothesis is that phosphenes are caused by spikes in a subset of excitatory projection neurons. In this case, pan-GABAergic stimulation would not be expected to produce phosphenes, but stimulation of specific GABAergic subtypes might. For example, stimulation of VIP-expressing neurons might produce a phosphene through disinhibition of excitatory neurons (*Cone et al., 2019*).

## Effect of laser power and optical fiber insertion on cortex

Over the course of this study, monkeys 2 and 3 acquired visual deficits in areas of the visual field corresponding to the regions of V1 inactivated. To ask whether the optogenetic manipulations produced long-lasting visual deficits, we conducted behavioral experiments in monkey 2 twenty-two months after the final optogenetic experiment was conducted (*Figure 8—figure supplement 2*). Visual sensitivity, assessed by the probability, latency, and accuracy of visually guided saccades, was reduced in the optogenetically manipulated lower-right visual field relative to the unmanipulated upper-left visual field, but the deficit was subtle. We presume that this deficit reflects cortical damage, which could be due to tissue heating by the light, repeated penetrations by optical fibers, or single-unit recordings that were made in this animal for three years prior to commencing the current study. While a comparable behavioral dataset could not be obtained from monkey 3, histological analysis of the calcarine sulcus, where most of the optogenetic manipulations were made in this animal, revealed nothing unusual (e.g. areas of necrosis, burn marks, etc.) (data not shown). They did reveal electrode/optical fiber tracks, the expected gliosis associated with these tracks, and healthy-looking mCherry+ neurons that were similar in morphology and density to those in monkey 1.

The laser power used in the current study spanned a broad range (4–160 mW for 200–300 ms) and, on average, was higher than that used in other studies (*Afraz et al., 2015*; *Cavanaugh et al., 2012*; *Dai et al., 2014*; *El-Shamayleh et al., 2017*; *Fetsch et al., 2018*; *Gerits et al., 2012*; *Inoue et al., 2015*; *Ohayon et al., 2013*; *Stauffer et al., 2016*; *Tamura et al., 2017*). Given the stimulation parameters we used, (450 nm light conducted through 300 μm-diameter optical fibers), we likely heated tissue near the fiber tip by several °C in many of our experiments (*Arias-Gil et al., 2016*). However, the consistency of the behavioral effect within individual sessions after repeated optical stimulation argues against acute damage (*Figure 8C–D*).

The tissue damage produced by optogenetic manipulations can be mitigated by using artificial dura (*Nandy et al., 2019*; *Ruiz et al., 2013*) and red-shifted or step-function opsins (*Berndt et al., 2009*). Artificial dura allows non-invasive optical stimulation of the superficial cortical layers, reducing mechanical damage. Red-shifted opsins are activated by long-wavelength light, which heats tissue less and thus causes less thermal damage than short-wavelength lights do. The neural activity produced by step-function opsins outlast the light pulses required to trigger them, allowing brief, safe light pulses to produce longer lasting stimulation events (*Gong et al., 2020*).

## Potential applications of AAV–mDlx5/6–ChR2

Optogenetic stimulation of inhibitory neurons using AAV–mDlx5/6–ChR2 facilitates at least three broad categories of experiments. The first category includes experiments in which slow neural inactivation precludes data collection, for example, experiments probing the neural substrate of life-sustaining processes (e.g. breathing) (*Baertsch et al., 2018*; *Simonyan, 2014*). Less extreme examples include the inactivation of oculomotor structures that are necessary for stable visual fixation, a simple oculomotor behavior without which more complicated behaviors are difficult to study (*Goffart et al., 2012*; *Krauzlis et al., 2017*). Experiments in which inactivation induces compensatory changes in task strategy (*Paolini and McKenzie, 1997*) or the routing of neural signals also fall in this category (*Cowey, 2010*; *Kinoshita et al., 2019*; *Leopold, 2012*; *Mori et al., 2006*).

The second class of experiments are those that address questions about the functional significance of spike timing. Monkeys can learn to use signals in sensory cortices at particular times relative to external and internal events to mediate their behavior (*Poort et al., 2012*; *Roelfsema et al., 1998*; *Seidemann et al., 1998*). Just as electrical microstimulation can be used to reveal the contribution of spikes added to sensory representations at specific times, optogenetic inactivation can be used, complementarily, to eliminate spikes. Indeed, optogenetic inactivation was used recently to show that spiking activity in the frontal eye fields of macaques contributes to memory-guided saccades before, during, and after target presentation (*Acker et al., 2016*). Future studies may reveal differences between the transient and sustained phases of sensory-, decision- and movement-related signals for guiding behavior (*Freedman et al., 2001*; *Hegdé, 2008*; *Ibos and Freedman, 2017*; *Roelfsema et al., 2007*; *Shushruth et al., 2018*).

A third class of experiments probes the electrophysiological response properties of inhibitory neurons in vivo (*Adesnik et al., 2012*; *Atallah et al., 2012*; *Cardin et al., 2009*; *Scholl et al., 2015*; *Sohal et al., 2009*; *Wilson et al., 2017*; *Wilson et al., 2012*). Excitatory and inhibitory neurons

within a cortical area have different response properties in mice, cats, and ferrets, a fact that is presumably related to differences in their respective functional contributions (*Huang and Paul, 2018*). Identification of inhibitory and excitatory neurons in vivo has been challenging in monkeys. The discovery of fast-spiking excitatory neurons in primates undermines the use of extracellularly recorded spike waveforms (*Kelly et al., 2019*). Optogenetic phototagging of inhibitory neurons, using AAV–mDlx5/6–ChR2 permits electrophysiological identification of neuronal subtypes more decisively (*Figure 9*).

In summary, the optogenetic approach used in this study holds promise for a finer level of neural circuit interrogation than previously achievable in monkeys. This union of neural inactivation technique and animal model has broad utility for addressing many outstanding questions in systems neuroscience that span the domains of sensation, action and cognition.

# Materials and methods

## Key resources table

| Reagent type (species) or resource | Designation | Source or reference | Identifiers | Additional information |
|---|---|---|---|---|
| Antibody | mCherry monoclonal antibody | Clontech | Cat. No. 632543 RRID:AB_2307319 | (1:250) |
| Antibody | Rabbit anti-parvalbumin | Swant | Code: 27 RRID:AB_2631173 | (1:5000) |
| Antibody | Donkey anti-Mouse IgG (H+L) highly cross-adsorbed secondary antibody, Alexa Fluor 568 | Molecular Probes | Cat. No. A10037 RRID:AB_2534013 | (1:200) |
| Antibody | Donkey anti-Rabbit IgG (H+L) highly cross-adsorbed secondary antibody, Alexa Fluor 488 | Molecular Probes | Cat. No. A21206 RRID:AB_2535792 | (1:200) |
| Antibody | Donkey anti-Rabbit IgG (H+L) highly cross-adsorbed secondary antibody, Alexa Fluor 350 | Molecular Probes | Cat. No. A10039 RRID:AB_2534015 | (1:200) |
| Other | DAPI | Invitrogen | Cat. No. D21490 | (1:5000) |
| Recombinant DNA reagent | pAAV-mDlx-ChR2-mCherry-Fishell-3 | Addgene | Cat. No. 83898 RRID:Addgene_83898 | |
| Cell line (Homo-sapiens) | HEK293T | American Type Culture Collection | CRL-3216 RRID:CVCL_0063 | |
| Biological sample (*Macaca Mulatta*) | Rhesus monkey | Washington National Primate Research Center | NA | |
| Software/ Algorithm | MATLAB | Mathworks | https://www.mathworks.com/products/matlab.html RRID:SCR_001622 | |
| Software/ Algorithm | Fiji | NIH (ImageJ) | https://imagej.net/Fiji RRID:SCR_002285 | |

*Continued on next page*

*Continued*

| Reagent type (species) or resource | Designation | Source or reference | Identifiers | Additional information |
|---|---|---|---|---|
| Software/ Algorithm | Plexon Sort Client | Plexon | http://www. plexon.com RRID:SCR_003170 | |
| Software/ Algorithm | Plexon Offline Sorter | Plexon | http://www. plexon.com RRID:SCR_000012 | |

## Contact for resource sharing

Further information and requests for resources should be directed to and will be fulfilled by the Lead Contact, Gregory D Horwitz (ghorwitz@u.washington.edu).

## Experimental model and subject details

Three rhesus monkeys (*Macaca mulatta*) participated in this study (males; 7–14 kg). Two monkeys were surgically implanted with a head-holding device and a recording chamber that provided access to the primary visual cortex (V1). Surgical procedures, experimental protocols, and animal care conformed to the NIH *Guide for the Care and Use of Laboratory Animals* and were approved by the Institutional Animal Care and Use Committee at the University of Washington.

Animal husbandry and housing were overseen by the Washington National Primate Research Center. All monkeys had ad-libitum access to biscuits (Fiber Plus Monkey Diet 5049, Lab Diet). Monkeys 2 and 3 had controlled daily access to fresh produce and water. When possible, animals were pair-housed and allowed grooming contact. Cages were washed every other week, bedding was changed every day, and animals were examined by a veterinarian at least twice per year.

During each experiment, monkeys viewed a CRT monitor binocularly with their heads fixed. The viewing distance was 100 cm for monkey 2 and 50 cm for monkey 3. Eye position signals were measured with an optical eye tracker for monkey 2 and a scleral search coil for monkey 3. Behavioral and stimulation timing events and eye position signals were digitized and stored for offline analysis.

## Method details

### AAV vector production

Recombinant AAV vectors were produced using a conventional three-plasmid transient transfection of human embryonic kidney cells (HEK293T, female, unauthenticated) with polyethylenimine (25 kDa, Polysciences). The transfer plasmid was pAAV-mDlx-ChR2-mCherry-Fishell-3 (Addgene #83898). Vectors were harvested and purified by ultracentrifugation through an iodixanol gradient column, exchanged into phosphate buffered saline (PBS), and titered using qPCR.

### AAV vector injections

After mapping a track through V1 gray matter using standard extracellular recording techniques in awake fixating monkeys, we advanced an electrode and cannula to the deepest point of the track and began a series of injections. Using a Hamilton syringe attached to a manual pump, we injected 1.0–1.5 µl of AAV vector at each of several locations spaced 500 µm apart along a track (normal to the opercular surface). Each injection was followed by a 2 min wait period after which the electrode and cannula were slowly retracted to the next site. This process was repeated at 9–14 sites, and a total of 14–17 µl was injected along each track. In monkey 2, we injected 14 µL of AAV9–mDlx5/6–ChR2–mCherry ($1.5 \times 10^{13}$ genomes/ml) at each of two opercular sites that were ~2 mm apart. The AAV vector was injected along 4 mm tracks throughout the thickness of the cortex at both sites, in the left hemisphere. In monkey 3, we injected ~17 µL of AAV1–mDlx5/6–ChR2–mCherry ($1.0 \times 10^{13}$ genomes/ml) along a 5 mm track in the first site and 14 µL along a 6.5 mm track in the second site, in the right hemisphere, to target both opercular and calcarine regions of area V1. The two injection tracks were ~1.5 mm apart.

## Histology

We injected area V1 of monkey 1 with AAV1–mDlx5/6–ChR2-mCherry to examine the specificity of vector transduction. These injections were performed during a surgical procedure while the monkey was anesthetized, and electrophysiological recordings were not made. The monkey recovered from the surgery and was euthanized 45 days later with an overdose of pentobarbital and perfused trans-cardially with 4% paraformaldehyde (wt/vol). The brain was removed, cryoprotected in 30% sucrose (wt/vol) and 50 μm-thick sections were cut on a sliding microtome. Fluorescence signals from mCherry (primary antibody: 1:250, Clontech 632543, mouse anti-mCherry; secondary antibody: 1:200, Invitrogen Molecular Probe) and parvalbumin (primary antibody: 1:5000, rabbit anti-PV, Swant 27; secondary antibody: 1:200, Invitrogen Molecular Probes) were detected immunocytochemically. Sections were counterstained with DAPI (1:5000, Molecular Probes D-21490) and cover-slipped using a DABCO-based mounting medium.

## Neurophysiology

Three to four weeks after AAV injections in monkeys 2 and 3, we searched for neuronal responses to blue light (450 nm; 33–161 mW) delivered to area V1 via an optical fiber (300 μm outer diameter; Thor Labs) with a beveled tip that eased entry through the dura. A fiber and a glass-coated tungsten electrode (1–3 MΩ FHC) were placed in a common guide tube and lowered independently into the brain by microdrive (Narashige or Alpha-Omega). Extracellular spikes were amplified (1x head-stage), high-pass filtered (250 Hz cutoff), digitized (sampling rate of 40 kHz) and sorted (Plexon MAP system).

## Site selection criteria

Stimulation sites were selected by inserting an electrode into V1 and finding a region with vigorous visual activity and a clearly defined receptive field (RF). The optical fiber was then lowered while repeatedly delivering brief laser pulses. The optical fiber typically lagged the electrode by 100–500 μm. Only sites at which optical stimulation produced an audible change in firing rate were tested.

## Laser setup

The laser was developed in-house by the Bioengineering Core at the Washington National Primate Research Center. Light was generated by a laser diode (part # PL TB450B). Light delivery was modulated by modulating the current to the laser diode (digital to analog converter part # AD5683) not by shutter.

## Visually guided saccade task

Monkeys were trained to make saccades to visual targets 4–17° from the fixation point. Each trial began when the monkey acquired a central fixation point (0.2–0.3° sided square) within a 1.6 × 1.6° electronic window. Then, 13 ms after the central target disappeared, a saccade target (square with sides 0.3–0.4°) was presented. Two to ten target locations, equiangularly spaced at fixed radius, were interleaved within each block of trials. Monkeys were rewarded for making a saccade to the target. On half of the target-present trials at each location, a 300-ms laser pulse was delivered simultaneously with the target presentation (*Figure 4B*). We interleaved 10–30 catch trials in which no target was presented, and the monkey was rewarded unconditionally. Optical stimulation was delivered on half of the catch trials, immediately after the fixation point disappeared.

## Two-alternative forced choice (2AFC) Gabor contrast detection task

Monkeys were trained to detect a Gabor stimulus positioned 4–17° from the fixation point. Each trial began when the monkey acquired the fixation point. Then, after a 520-ms delay, a drifting Gabor stimulus appeared on half of the trials (spatial frequency = 1 cycle/°, temporal frequency = 8 Hz, standard deviation = 0.2°, duration = 200 ms). Immediately after the Gabor stimulus disappeared, a pair of targets appeared along the horizontal meridian, 2° from the fixation point. A saccade to the target on the same side of the screen as the Gabor stimulus was rewarded on Gabor-present trials, and a saccade to the target on the opposite side was rewarded on Gabor-absent trials.

The Gabor stimulus appeared inside the RF of neurons at the recording site in all trials except a few in which retinotopic specificity was tested (*Figure 6*). Optical stimulation began at the stimulus

onset, lasted 300 ms (*Figure 5B*), and was delivered on half of the Gabor-present and half of the Gabor-absent trials. The monkey typically performed several blocks of trials per session, each consisting of 120 trials. Stimulus strength was adjusted by independent contrast staircase procedures on laser and control trials. Contrast, defined as the difference between the highest and lowest luminance values, divided by the sum of the two, increased by a factor of 1.18–1.33 following an incorrect response and decreased by a factor of 0.75–0.85 following three consecutive correct responses.

## Quantification and statistical analyses

All statistical analyses were performed in MATLAB.

### Sensitivity index (d')

Sensitivity (*d'*) was measured using a standard formula from signal detection theory (*Green and Swets, 1966*; *Macmillan and Creelman, 2004*).

$$d' = \Phi^{-1}(proportion\,of\,hits) - \Phi^{-1}(proportion\,of\,false\,alarms)$$

In this equation, $\Phi^{-1}$ is the inverse normal cumulative distribution function. Proportions of 0 were replaced with 0.5/n, and proportions of 1 were replaced by 1-0.5/n, where n is the number of Gabor-present (for hits) or Gabor-absent trials (for false alarms) (*Stanislaw and Todorov, 1999*).

### Fit to behavioral data

Proportions of correct responses were fit with a cumulative Weibull distribution function by maximizing likelihood assuming binomial error. Fitting was performed using the inbuilt MATLAB *fmincon* function. Detection threshold was defined as the luminance contrast corresponding to 82% correct.

## Acknowledgements

We thank Michael N Shadlen for helpful comments on the manuscript, Elizabeth Buffalo for generous microscope access, and Albert Ng for help with MRI-related software. This work was funded by NIH EY018849 to Gregory D Horwitz, NIH/ORIP grant P51OD010425 to Washington National Primate Research Center, NEI Center Core Grant for Vision Research P30 EY01730 to the University of Washington and R90 DA033461 (Training Program in Neural Computation and Engineering) to Abhishek De.

## Additional information

### Funding

| Funder | Grant reference number | Author |
| --- | --- | --- |
| National Eye Institute | EY030441 | Gregory D Horwitz |
| NIH Office of Research Infrastructure Programs | P51OD010425 | Abhishek De<br>Yasmine El-Shamayleh<br>Gregory D Horwitz |
| National Institutes of Health | R90 DA033461 | Abhishek De |
| National Eye Institute | P30 EY01730 | Abhishek De<br>Yasmine El-Shamayleh<br>Gregory D Horwitz |

The funders had no role in study design, data collection and interpretation, or the decision to submit the work for publication.

### Author contributions

Abhishek De, Yasmine El-Shamayleh, Data curation, Formal analysis, Investigation; Gregory D Horwitz, Conceptualization, Supervision, Project administration

## Author ORCIDs
Abhishek De https://orcid.org/0000-0002-2978-473X
Yasmine El-Shamayleh https://orcid.org/0000-0002-5396-2823
Gregory D Horwitz https://orcid.org/0000-0001-5130-5259

## Ethics
Animal experimentation: Surgical procedures, experimental protocols and animal care conformed to the NIH Guide for the Care and Use of Laboratory Animals and were approved by the Institutional Animal Care and Use Committee at the University of Washington (IACUC protocol #4167-01).

## Decision letter and Author response
Decision letter https://doi.org/10.7554/eLife.52658.sa1
Author response https://doi.org/10.7554/eLife.52658.sa2

## Additional files

### Supplementary files
• Transparent reporting form

### Data availability
All data have been uploaded to https://github.com/horwitzlab/fast-and-reversible-neural-inactivation (copy archived at https://github.com/elifesciences-publications/Fast-and-reversible-neural-inactivation).

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
