## [Decision Letter]

**Acceptance summary:**

Targeted optogenetic inactivation of neural circuits in non-human primates is essential to clarify specific links between neuronal activity and behaviour. Here the authors capitalise on the recent development of Dlx5/6 enhancer-guided targeting of GABAergic neurons (Dimidschstein et al., 2016) for optogenetic manipulation of macaque primary visual cortex (V1). The authors show how optogenetic targeting of V1 GABAergic neurons modulates V1 neural activity and leads to a substantial, specific impairment in vision guided behaviour.

**Decision letter after peer review:**

Thank you for submitting your article "Fast and reversible neural inactivation in macaque cortex by optogenetic stimulation of GABAergic neurons" for consideration by *eLife*. Your article has been reviewed by three peer reviewers, including Michael Schmid as the Reviewing Editor and Reviewer #1, and the evaluation has been overseen by Joshua Gold as the Senior Editor. The following individuals involved in review of your submission have agreed to reveal their identity: Wim Vanduffel (Reviewer #2); David Sheinberg (Reviewer #3).

The reviewers have discussed the reviews with one another and the Reviewing Editor has drafted this decision to help you prepare a revised submission.

Summary:

De, El-Shamayleh, and Horwitz describe anatomical, electrophysiological and behavioral results of optogenetic deactivation experiments targeting primary visual cortex (V1) in macaques. The authors capitalise on the recent development of Dlx5/6 enhancer-guided targeting of GABAergic neurons (Dimidschstein et al., 2016). Here De et al. used the Dlx5/6 enhancer in combination with a depolarizing opsin (ChR2) to activate inhibitory neurons, with the aim to inactivate the local downstream excitatory neurons. A key advance of this study is the demonstration that a AAV approach for targeting inhibitory neurons that has been shown to work in the marmoset translates to the rhesus monkey. The authors show histological evidence for transduced neurons near the injection site, as evidenced by mCherry expression. Moreover, most of the transduced neurons were PV+, indicating high specificity for inhibitory neurons. At neuron level, they observed both increased (2/3 of the stimulated sites) and suppressed (1/3 of the sites) light-induced activity. Moreover, the monkeys failed to make reliable saccades to targets represented by the stimulated neurons. Finally, the monkeys had severely reduced contrast detection thresholds at these sites. The authors provide compelling results from a combination of histological, electrophysiological and behavioural tests. Particularly the strong behavioural effects advance the field and will be of great interest to a wide audience. Unwanted rebound effects, which are typically present when using alternative hyperpolarizing opsins (e.g. ArchT or Jaws), are largely absent. Overall, the evidence presented is solid, the analyses are sound, the writing is very straightforward and the message is clear. The new research is important, timely and provides an important step forward for the field. However, the reviewers have expressed important concerns that need to be addressed before the manuscript can be accepted for publication.

Essential revisions:

1) The expression pattern needs to be more fully characterised. The selectivity of the vector seems to be high -i.e. mainly restricted to PV+ interneurons. Yet, the sensitivity is surprisingly low (only 41 neurons are transduced). Would this be a vast underestimation of the real number of the neurons expressing ChR2? It would be good for readers to know the percentage of Parvalbumin neurons that show fluorescence to get a better estimate on the expression sensitivity. There are some recent reports indicating that the threshold for detecting FP expression might be higher than the threshold for the functional gene (Kinoshita et al., 2019). Or do the authors think that the number of neurons expressing ChR2 can be as low as ~40 in order to evoke a clear behavioral effect? Figure 1 suggests a very laminar-specific expression pattern, but the authors explain that this is not typical. Was this slice the only one analyzed? Ideally the reader would like to see an assessment across cortical laminae, but perhaps the authors could show further sections that give a more representative view of the expression pattern. Although seemingly annoying, this may be useful for layer-specific optogenetic deactivations.

2) Characterise more fully electrophysiological responses. Given the relatively long latencies of the optogenetic effect (see Figure 2—figure supplement 2), it is unlikely that these are only first order neurons expressing the opsin which are directly activated by the blue light. How do the authors explain the long latency effects? It would be also interesting to plot the latencies of the cells showing a suppression effect (i.e. the time after stimulation onset that the activity drops significantly below the pre-stimulation firing rate). These latencies should be longer than those of neurons showing an excitatory effect. Estimating the onset of suppression is not trivial, but this could be informative regarding potential direct and indirect effects. Figure 3B also shows suppression for a site with some very bursty responses which seem to drastically inflate the Y-axis (Response). Was this high variability and bursty activity common for suppressed sites? The overall spontaneous rates of many GABAergic cells is fairly high, but it's not clear if that is the case for the population explored here. That spiking increased in >60% of recorded sites is in line with successful targeting of GABAergic interneurons. But what do we learn about these neurons? The authors discuss the potential of photo-tagging in the Discussion and provide one exemplary direction selective unit in Figure 9. But one is left with the question what happened at the other sites? Are they visually responsive?

3) Further aspects should be considered that might have influenced behavioural performance. For the behavioral tasks the authors should probably emphasize that reward contingencies were not dependent on laser delivery. It was also unclear on why the measure used quantifying the effect for the oculomotor task was not simply distance from the target? For this task, the data for Monkey 3 shown in Figure 4 even for the control trials looks like it's not right on the center of the RF location. Does this figure show exactly where the target was presented and how were the eye positions calibrated? In both paradigms opto stimulation occurred at the same time as visual stimulation. Given a visual response latency of 40 ms or more in V1 neurons, at least in theory, the opto stimulus could serve as a cue telling the monkey how to act in order to get reward. It is indicated that the change in contrast detection performance is due to the reduction in sensitivity and not a change in criterion. One cannot conclude that from d-primes only. The c-criterion should also be listed as there can be a change in sensitivity and criterion.

4) Electrophysiological and behavioural measures should be more directly related to each other. There's no obvious reason why these couldn't be done simultaneously. If possible, it would be good to see opto elicted spiking activity from the trials during behavioural testing and to probe whether there is a direct relationship between the strength of spiking and the behavioural effect.

5) Clarify for the detection conditions, how the authors move from the example sessions (Figure 5) to the population data (Figure 7). Some rewording here to make it clear that the comments in the paragraph below are referring to the examples in Figure 5 and not the whole population (which follows in a couple of paragraphs). For the population, the authors should revisit Figure 7 to not include all the blocks, as this conflates the independent sessions (11 and 12) from the blocks, which are clearly not independent. To include all the blocks in Figure 7 is a clear case for pseudo-replication. The population analysis needs to be by session, not block. The authors should also revisit the psychometric fits (examples in Figure 5, e.g.). The laser fits don't look very good – was there some estimate of goodness of fit for these?

6) Clarify details about injection and stimulation procedures (see minor points), including heating induced damage considerations. A concern is in understanding and justifying the need for the large increase in power used to activate the neurons under study. The absolute power levels are on a direct concern if they cause lasting damage to the tissue. On one hand the prolonged efficacy across the session is evidence that effects of greater power did not present an acute problem, but there could be concern that prolonged use in a single site, for example, could lead to irreversible damage. More discussion on the power would be useful.

---

## [Author Response]

Essential revisions:1) The expression pattern needs to be more fully characterised. The selectivity of the vector seems to be high -i.e. mainly restricted to PV+ interneurons. Yet, the sensitivity is surprisingly low (only 41 neurons are transduced). Would this be a vast underestimation of the real number of the neurons expressing ChR2? It would be good for readers to know the percentage of Parvalbumin neurons that show fluorescence to get a better estimate on the expression sensitivity.

The reviewers are correct that the number of transduced neurons in Figure 1 is a vast underestimate. Indeed, many more neurons expressed ChR2 in monkey 1 than are shown in the Figure 1. Unfortunately, a nearby injection of an entirely different viral vector that expressed ChR2-eYFP under the control of a different promoter complicated our analyses of selectivity and sensitivity. Transduction by the two vectors is easily distinguished on the basis of their distinct fluorescent protein genes. However, the spectral overlap between the eYFP signal and the green secondary antibody we used to label PV neurons required us to look at sections where the two injections did not overlap. In Figure 1, we show a section of V1 near the edge of the AAV-mDlx5/6-ChR2mCherry transduction zone that lacks ChR2-eYFP expression; this region allows us to estimate selectivity easily but provides an underestimate of sensitivity due to the sparse mCherry label.

To address reviewers’ comments, we have analyzed a substantially larger region in monkey 1 that spans the V1–V2 border (Figure 1—figure supplement 1). We have amplified the AAV-mDlx5/6-ChR2-mCherry signal and PV signal, the latter using a short wavelength secondary antibody that avoids confusion with the ChR2-eYFP signal from the second viral vector. This section had many more ChR2-mCherry+ cells (N=543). In regions of efficient transduction, we estimate that ~50% of parvalbumin+ neurons were transduced.

There are some recent reports indicating that the threshold for detecting FP expression might be higher than the threshold for the functional gene (Kinoshita et al., 2019). Or do the authors think that the number of neurons expressing ChR2 can be as low as ~40 in order to evoke a clear behavioral effect?

The threshold for detecting FP expression may indeed exceed the threshold for functional ChR2 expression. We were able to detect native FP signal in histological sections from monkey 1, suggesting that ChR2 expression was likely sufficient to manipulate spiking activity (monkey 1 was not used in the electrophysiological/ behavioral experiments). To facilitate cell counting, all sections shown in the manuscript were amplified immunohistochemically.

The minimum number of V1 neurons that must be manipulated to cause a behavioral effect is an important issue that our data do not speak to. The number of neurons affected by the light stimulation may depend on the efficiency of AAV transduction, the shape of the optical fiber tip, spread of the laser light, tissue transmissibility, laser power, sensitivity of the behavioral assay, and other factors.

Figure 1 suggests a very laminar-specific expression pattern, but the authors explain that this is not typical. Was this slice the only one analyzed? Ideally the reader would like to see an assessment across cortical laminae, but perhaps the authors could show further sections that give a more representative view of the expression pattern. Although seemingly annoying, this may be useful for layer-specific optogenetic deactivations.

The efficiency of transduction across cortical laminae is determined by in part by where and how the vector injection is made. The AAV injections into monkey 1 were made during a surgical procedure, without electrophysiological guidance, which may explain the concentration of expression in superficial cortical layers. Injections into monkeys 2 and 3 were based on electrophysiological depth measurements and are therefore more likely to have spanned all V1 layers. To provide the reviewers with evidence for expression in deeper layers, we have recently made another injection of AAV-mDlx5/6ChR2-mCherry into area V4 of a monkey that was not used in this study. In that experiment, transduction spanned all of the layers (except for layer 4 which, in our hands, is difficult to transduce efficiently irrespective of the vector injected). Please see Figure 1B of http://www.pnas.org/content/116/52/26195 for the results of that experiment.

2) Characterise more fully electrophysiological responses. Given the relatively long latencies of the optogenetic effect (see Suppl Figure 1), it is unlikely that these are only first order neurons expressing the opsin which are directly activated by the blue light. How do the authors explain the long latency effects?

The long latency effects are likely due to complex network activity within and beyond V1. The existence of these complex interactions means that selective optical stimulation of inhibitory neurons need not necessarily exert a net-inhibitory effect on the circuit.

It would be also interesting to plot the latencies of the cells showing a suppression effect (i.e. the time after stimulation onset that the activity drops significantly below the pre-stimulation firing rate). These latencies should be longer than those of neurons showing an excitatory effect. Estimating the onset of suppression is not trivial, but this could be informative regarding potential direct and indirect effects.

We agree with the reviewers that estimating the onset of suppression is not trivial. Decreases in firing rate are more difficult to detect than increases in firing rate, especially when the baseline firing rate is low, as is often the case in V1. We have done our best to estimate the latency of the optogenetic effect for both activated and suppressed sites. As anticipated by the reviewer, the latencies at suppressed sites were longer than those at activated sites (Figure 2—figure supplement 2C). However, the interpretation of this result is complicated; the expected delay from synaptic transmission is brief relative to the bias produced by estimating a reduction in an already-low firing rate (relative to an increase). We have provided raster plots that i l lust rate laser responses at ever y suppressed site we studied (Figure 2—figure supplement 3).

Figure 3B also shows suppression for a site with some very bursty responses which seem to drastically inflate the Y-axis (Response). Was this high variability and bursty activity common for suppressed sites? The overall spontaneous rates of many GABAergic cells is fairly high, but it's not clear if that is the case for the population explored here.

Suppressed sites were not unusually variable and bursty. The example neuron shown in Figure 3B was selected specifically because its baseline firing rate was high (it also happened to be bursty), which made the suppression effect particularly clear. Most suppressed sites had low baseline firing rates, making suppression less obvious (Figure 2—figure supplement 3). The baseline firing rates of suppressed and activated sites were similar (p=0.87; unpaired t test).

That spiking increased in >60% of recorded sites is in line with successful targeting of GABAergic interneurons. But what do we learn about these neurons? The authors discuss the potential of photo-tagging in the Discussion and provide one exemplary direction selective unit in Figure 9. But one is left with the question what happened at the other sites? Are they visually responsive?

All neurophysiological and behavioral data were collected concurrently except for the data in Figure 9, which was collected during a block of fixation trials. Once we found a site that was modulated by the laser, we focused on documenting the behavioral deficit. The non-stationarity of firing rate apparent in a few of the plots in Figure 2—figure supplement 3 is due to changes in isolation quality.

At most sites—both activated and suppressed by the laser—presentation of the Gabor stimulus evoked a response (Figure 2—figure supplement 1). Of the 56 sites, 46 had elevated responses during visual stimulation, and of those, 19 attained statistical significance (p<0.05, Mann Whitney U test). We report these numbers in the revised manuscript. The weakness of the visual response is expected. We did not tailor the visual stimulus (an achromatic, 1 cycle/° upward-drifting, hor i zontal Gabor pat tern) to the preferences of the neurons at the stimulation site, and the contrast of the Gabor stimulus was usually low because it was adjusted by a staircase procedure to be near psychophysical detection threshold.

3) Further aspects should be considered that might have influenced behavioural performance. For the behavioral tasks the authors should probably emphasize that reward contingencies were not dependent on laser delivery.

We emphasize in the revised manuscript that the reward contingencies were not dependent on laser delivery.

It was also unclear on why the measure used quantifying the effect for the oculomotor task was not simply distance from the target?

Thank you for the suggestion. We have repeated the analysis of saccade-task performance using distance from the end point to the target as suggested by the reviewer (Figure 4—figure supplement 1).

For this task, the data for Monkey 3 shown in Figure 4 even for the control trials looks like it's not right on the center of the RF location. Does this figure show exactly where the target was presented and how were the eye positions calibrated?

We have represented the target locations outside RFs in the revised figure (Figure 4). The figures show the nominal locations of the targets on the screen and calibrated estimates of eye position relative to these locations. Our eye position calibration is imperfect but is reasonably accurate (< 1° error).

Monkey 3 made inaccurate saccades into the left visual field even on some control trials (Figure 4—figure supplement 1, Figure 4E–F). This was likely due to repeated electrode penetrations into the midbrain of this animal, unrelated to the current experiments, that resulted in oculomotor deficits. This animal exhibited a leftward nystagmus that precluded accurate fixation behavior several months before the collection of data presented in this manuscript. During data collection for the current study, this animal developed several blind spots presumably due to cortical damage. The nystagmus is unrelated to the optogenetic manipulations made in this study and therefore unlikely to be of interest to the readers of *eLife*, but the blind spots are relevant and now discussed in the revised Discussion.

In both paradigms opto stimulation occurred at the same time as visual stimulation. Given a visual response latency of 40 ms or more in V1 neurons, at least in theory, the opto stimulus could serve as a cue telling the monkey how to act in order to get reward.

We agree with the reviewers, and have elaborated on this point in the revised manuscript. Indeed, if the monkey had been able to detect the optical stimulation, he might have been able to use this information in the saccade task to get reward on optical stimulation trials (Figure 4, Figure 4—figure supplement 1A–B). On trials in which optical stimulation was delivered and no target was visible, a saccade into the receptive fields of the stimulated neurons would often have been rewarded. The fact that the monkey did not routinely make saccades to the target in the RFs of the illuminated neurons suggests that he was unable to detect the stimulation, or at least was unable to use it to direct his saccades. In the contrast detection task, the optical stimulation does not provide a cue that is useful for getting a reward. The two possible choices are equally likely to be rewarded on both control and laser stimulation trials.

It is indicated that the change in contrast detection performance is due to the reduction in sensitivity and not a change in criterion. One cannot conclude that from d-primes only. The c-criterion should also be listed as there can be a change in sensitivity and criterion.

We now address this point in the revised manuscript. We can explain the changes in c-criterion and d’ using a model in which the effect of the laser is to make the signal distribution more similar to the noise distribution, and we include a figure for the reviewers illustrating this point (Figure 7—figure supplement 1). We are unable, however, to explain the changes in *d’* on the basis of a change in subjective criterion alone; a pure change in criterion does not affect *d’*. We cannot rule out the possibility that the laser changes the monkeys' subjective criterion and sensitivity, but the brevity and unpredictability of the optical stimulation argues against a large change in criterion.

4) Electrophysiological and behavioural measures should be more directly related to each other. There's no obvious reason why these couldn't be done simultaneously. If possible, it would be good to see opto elicted spiking activity from the trials during behavioural testing and to probe whether there is a direct relationship between the strength of spiking and the behavioural effect.

All of the electrophysiological recordings, with the exception of those in Figure 9, were made while the monkeys were performing the contrast detection task. We clarify this point in the revised manuscript.

We looked for a relationship between the strength of optical stimulation on spiking responses and behavioral effects, and we observed a weak, positive correlation that failed to reach statistical significance (Figure 8—figure supplement 1B).

In interpreting this result, it is important to consider uninteresting reasons for finding such a relationship and also for not finding one. The laser power changed across blocks of trials. Low laser power affects neural responses and behavior weakly, and high laser power affects both strongly, which would be expected to produce a positive correlation. A reason for not detecting such a correlation is that the electrodes recorded only a fraction of the neurons affected by the optical stimulation, and the quality of the neural signal varied from day to day. These two sources of variability (laser power and recording quality), prevent us from accurately estimating the relationship between the electrophysiologically generated response (across all neurons) and the resultant behavioral effect.

Nevertheless, in one session, we manipulated the laser power on seven blocks of trials, keeping the fiber position, electrode position and the spatial location of stimulus fixed. Under these conditions, we were able to observe a clear, positive correlation between neural and behavioral modulation (Figure 8—figure supplement 1A).

5) Clarify for the detection conditions, how the authors move from the example sessions (Figure 5) to the population data (Figure 7). Some rewording here to make it clear that the comments in the paragraph below are referring to the examples in Figure 5 and not the whole population (which follows in a couple of paragraphs).

We have rewritten the confusing passages to clarify the division between the example data in Figure 5 and the population data in Figure 7.

For the population, the authors should revisit Figure 7 to not include all the blocks, as this conflates the independent sessions (11 and 12) from the blocks, which are clearly not independent. To include all the blocks in Figure 7 is a clear case for pseudo-replication. The population analysis needs to be by session, not block.

We have repeated the analysis in Figure 7, treating each session as an independent observation.

The authors should also revisit the psychometric fits (examples in Figure 5, e.g.). The laser fits don't look very good – was there some estimate of goodness of fit for these?

We agree that the fits to the psychometric function data on laser trials are not very good. There are two reasons for this. First, in many blocks, performance increased shallowly with stimulus contrast because of the strong inactivation. Performance on laser trials even at the highest contrast was therefore poor, forcing the psychometric fit to have a shallow slope within the range of contrasts tested. Second, not all stimulus contrasts were probed equally often because of the staircase procedure. This fact can give the appearance of a poor model fit. The fit takes into account the number of stimulus presentations at each contrast. As a result, the model more accurately fits the points that were probed more often. We have replotted the psychometric functions, representing the number of stimulus presentations at each contrast as the size of the corresponding data points (Figures 5–6, Figure 6—figure supplement 1).

The deviance, the measure of fitting error that is minimized in generalized linear models, was actually lower on control trials (median = -20.07) than on laser trials (median = -17.76) because of the flatness of the psychometric function over the range of contrasts we were able to test.

6) Clarify details about injection and stimulation procedures (see minor points), including heating induced damage considerations. A concern is in understanding and justifying the need for the large increase in power used to activate the neurons under study. The absolute power levels are on a direct concern if they cause lasting damage to the tissue. On one hand the prolonged efficacy across the session is evidence that effects of greater power did not present an acute problem, but there could be concern that prolonged use in a single site, for example, could lead to irreversible damage. More discussion on the power would be useful.

We have clarified the details of the injection and stimulation procedures in the revised manuscript.

During some of our initial experiments, we used high laser power because we did not know a priori the laser power needed to induce a behavioral effect. However, we show that a laser power as low as 30 mW is sufficient to achieve a strong behavioral effect (Figure 8B).

Both monkeys currently have scotomas in areas of the visual field corresponding to some of the regions of V1 inactivated. The laser power used in some experiments was unnecessarily high and likely caused thermal damage in the stimulated regions. As pointed out the reviewer, we did not observe any acute change in the behavioral effect over repeated stimulation (Figure 8C–D) but we cannot rule out that heating did not lead to permanent damage. Another likely cause of permanent damage in these experiments is mechanical due to repeated insertions of the 300 µm optical fibers. We have added discussion on these points and potential remedies for future experiments in the revised manuscript.